# BinauralGrad: A Two-Stage Conditional Diffusion Probabilistic Model for Binaural Audio Synthesis

**Yichong Leng**[1*†]**, Zehua Chen**[3*]**, Junliang Guo**[2]**, Haohe Liu**[5]**, Jiawei Chen**[6]**, Xu Tan**[2]
**Danilo Mandic**[3]**, Lei He**[4]**, Xiang-Yang Li**[1]**, Tao Qin**[2]**, Sheng Zhao**[4]**, Tie-Yan Liu**[2]
[1]University of Science and Technology of China, [2]Microsoft Research Asia
[3]Imperial College London,[4]Microsoft Azure Speech,[5]University of Surrey
[6]South China University of Technology
[1]`lyc123go@mail.ustc.edu.cn,xiangyangli@ustc.edu.cn`
[2]`{junliangguo,xuta,taoqin,tyliu}@microsoft.com`
[3]`{zehua.chen18,d.mandic}@imperial.ac.uk`
[4]`{helei,szhao}@microsoft.com`
[5]`hl01486@surrey.ac.uk,`[6]`csjiaweichen@mail.scut.edu.cn`

`https://github.com/microsoft/NeuralSpeech`

## Abstract

Binaural audio plays a significant role in constructing immersive augmented and virtual realities. As it is expensive to record binaural audio from the real world, synthesizing them from mono audio has attracted increasing attention. This synthesis process involves not only the basic physical warping of the mono audio, but also room reverberations and head/ear related filtrations, which, however, are difficult to accurately simulate in traditional digital signal processing. In this paper, we formulate the synthesis process from a different perspective by decomposing the binaural audio into a common part that shared by the left and right channels as well as a specific part that differs in each channel. Accordingly, we propose BinauralGrad, a novel two-stage framework equipped with diffusion models to synthesize them respectively. Specifically, in the first stage, the common information of the binaural audio is generated with a single-channel diffusion model conditioned on the mono audio, based on which the binaural audio is generated by a two-channel diffusion model in the second stage. Combining this novel perspective of two-stage synthesis with advanced generative models (i.e., the diffusion models), the proposed BinauralGrad is able to generate accurate and high-fidelity binaural audio samples. Experiment results show that on a benchmark dataset, BinauralGrad outperforms the existing baselines by a large margin in terms of both object and subject evaluation metrics (Wave L2: 0.128 vs. 0.157, MOS: 3.80 vs. 3.61). The generated audio samples[3] and code[4] are available online.

## 1 Introduction

Human brains have the ability to decode spatial properties from real-world stereophonic sounds, which helps us to locate and interact with the environment. While in artificial spaces such as augmented and virtual reality, generating accurate binaural audio from the mono one is therefore essential to provide

---

[*]Equal contribution.

[†]This work was conducted at Microsoft. Corresponding author: Xu Tan, xuta@microsoft.com

[3]`https://speechresearch.github.io/binauralgrad`

[4]`https://github.com/microsoft/NeuralSpeech/tree/master/BinauralGrad`

36th Conference on Neural Information Processing Systems (NeurIPS 2022).

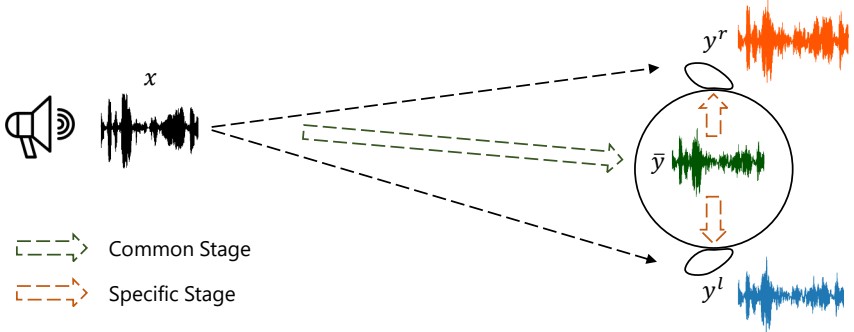

Figure 1: An illustration of the proposed two-stage framework. $x$ indicates the mono audio emitted by the source object, $y = (y^l, y^r)$ indicate the binaural audio received by the left and right ears of the receiver, and $\bar{y} = \text{mean}(y^l, y^r)$ is the mono audio calculated by averaging the two channels in $y$. The common stage models the factor affecting both the left and right channel such as the distance of source and listener and the interaction of audio with room and head. And the specific stage models the difference of left ear and right ear including the marginal distance difference and the difference of impulse response of two ears.

immersive environments for listeners. Traditional Digital Signal Processing (DSP) systems generate binaural audio by warping the mono audio according to the time difference between two ears and modeling room reverberations (e.g., room impulse response (RIR)) and head/ear related filtration (e.g., head-related transfer functions (HRTF)) through a linear time-invariant system [32, 47, 35]. However, they fail to generate accurate and immersive binaural audio due to the non-linear nature of sound propagation and the simplification in DSP systems, such as simplifying the physical modeling of RIR and utilizing general HRTFs instead of personalized. In addition, the exact physical information of the recording environment is not available most of the time.

Instead of following the complicated synthesis process of binaural audio in traditional digital signal processing which is difficult to model, in this paper, we formulate the process from a different and novel perspective. We first decompose the synthesis of binaural audio into two steps: 1) the original mono audio is emitted by the object and then diffuses to the listener. The audio that arrives at the left and right ears are similar, as the distance between the two ears is marginal compared with that between the object and the listener; 2) the binaural audio is then encoded by human head depending on the marginal difference of audio received at two ears [2, 43]. This process is personalized as the morphology of each listener is unique, which will change the internal encoding results. Precisely processing the two stages is difficult due to the complexity of physical system. Therefore, as an alternative, given that the two channels of the binaural audio is similar but with marginal differences, we propose to divide the the representation of the binaural audio $y = (y^l, y^r)$ as two parts. One is the general part that contains the common information of two channels, which are represented as the average $\bar{y}$; the other one is the specific part that is distinct between the two channels encoded by human head in the end, which are denoted as $\delta^l$ and $\delta^r$. Then we can represent the binaural audio as $y^l = \bar{y} + \delta^l$ and $y^r = \bar{y} + \delta^r$ respectively. The common part represents the fundamental information of the source mono audio, while the specific parts contain the difference between the left and right channels caused by room reverberance and the acoustical influence of human head.

Based on the representation, we propose a two-stage framework to model the two parts respectively. The common information of the two channels of binaural audio is generated in the first stage, and their difference are generated in the second stage. We therefore term them as first/common and second/specific stages. The common stage is encouraged to generate $\bar{y}$ while the specific stage tries to model the difference to the left and right audio. Specifically, we build our framework on denoising diffusion probabilistic models (diffusion models for short) [12], which have been shown effective and efficient to generate high-fidelity audio on the task of speech synthesis [15, 4, 17]. In the common stage, the mono audio is taken as the condition of a single-channel diffusion model, which is supervised by the average over the two channels of the golden binaural speech $\bar{y}$, encouraging the model to generate the common information of two channels. And in the specific stage, conditioned on the single-channel output of the first stage, we utilize a two-channel diffusion model to generate the audio for the left and right ears respectively. We provide an illustration of the proposed framework in Figure 1.

The main contributions of this work are summarized as follows:

- We formulate the representation of binaural audio as two parts, and propose a novel two-stage framework to respectively synthesize their common and specific information conditioned on the mono audio.
- Equipped with denoising probabilistic diffusion models, the proposed two-stage framework is able to generate accurate and high-fidelity binaural audio.
- We conduct experiments on the benchmark dataset released in [29], and the proposed framework outperforms existing baselines by a large margin, both on automatic (Wave L2: 0.128 vs. 0.157) and human evaluation (MOS: 3.80 vs. 3.61) metrics.

## 2 Related Works

### 2.1 Binaural Audio Synthesis

Different from text to speech synthesis that mainly generates mono speech from text [37, 38], binaural audio synthesis aims to convert mono audio into its binaural version. Based on the physical process of sound rendering, human listening can be generally considered as a source-medium-receiver model [3]. The sound waves emitted by the object will travel through the medium, i.e., free air and then be received by listener. In medium, the sound will be diffused, reverberated and effected by the physical objects such as walls during the propagation. Room impulse response (RIR) [19, 36, 1, 30] describes the distortion effect caused by surrounding environment using filters. Besides the interaction with medium, the morphology of listener (torso, head, and pinna) will also change the sound received by the eardrum. The head-related transfer functions (HRTF) [3, 6] are utilized to describe the transformation of the sound from a direction in free air to that it arrives at the eardrum [18]. Digital Signal Processing (DSP) methods synthesize the binaural speech by combining RIR and HRTF [47, 35], both of which are expensive and tedious to measure, as well as missed from most open-sourced datasets in addition. Therefore, DSP methods usually utilize generic functions instead, resulting in sub-optimal generation results as the recording environments are highly specialized among different datasets.

Recently, neural rendering approaches are proposed for binaural speech synthesis. Gebru et al. [10] utilizing neural networks to learn HRTFs implicitly. Richard et al. [29] propose a convolution neural network based model with an additional neural time warping module to learn the time shifts from the mono to binaural audio. Some works incorporate extra modalities such as the visual information to help the synthesis of binaural speech [46, 45, 26]. In this work, we consider the most basic scenario where only the mono audio is available.

### 2.2 Denoising Diffusion Probabilistic Models

Denoising diffusion probabilistic models (diffusion models for short) have achieved the state-of-the-art (SOTA) generation results in various tasks, including image [34, 22, 8, 7, 33, 39, 44] and super resolution image generation [13, 31, 41, 25], text-to-image generation [23, 11, 14, 28], text-to-speech synthesis [4, 15, 27, 17, 16, 5] and speech enhancement [20, 21, 42]. Especially, in audio synthesis, diffusion models have shown strong ability in modelling both spectrogram features [27, 17] and raw waveforms [4, 15, 5].

Diffusion models require a large number of inference steps to achieve high sample quality. For high-resolution data generation, some works [8, 13, 28] even cascade multiple diffusion models together and achieve the SOTA quality in text-to-image generation [28]. In this cascaded structure, multiple diffusion models are trained at different resolution scales, and while inference, low-resolution data samples are firstly generated with the base diffusion model, and then stacked conditional diffusion models are employed for super resolution, where high-fidelity results can be achieved with moderate time cost.

In this paper, for generating binaural audio waveform with high resolution (48kHz sampling rate), we integrate diffusion models into the proposed two-stage framework, and generate the common and specific information of binaural audio with single- and two-channel diffusion models respectively.

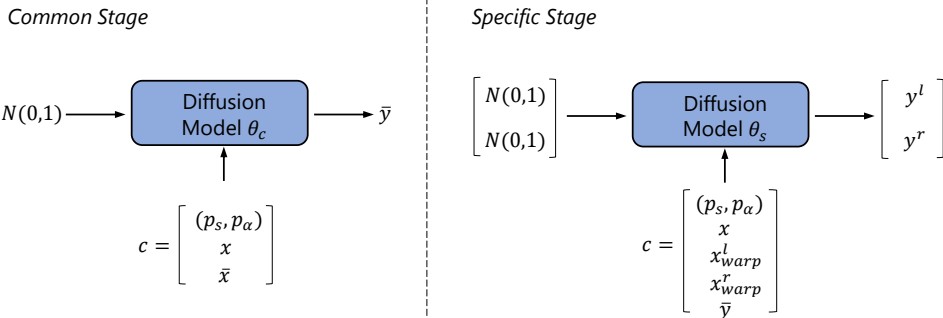

Figure 2: The pipeline of the proposed framework while training. The difference between the two stages includes the condition signal $c$, the parameter set $\theta$, and utilizing one-/two-channel diffusion models in common/specific stages respectively.

## 3 Methods

We introduce the proposed framework BinauralGrad in this section. We start with the preliminary knowledge about binaural audio and geometric warping, and then introduce the proposed two-stage framework as well as the structure of the diffusion models.

### 3.1 Preliminary

**Problem Definition**  Given the source mono audio $x \in R^N$ with length $N$ and the relative position between the source and listener $p$, we aim to generate the binaural audio $y = (y^l, y^r)$ that contains two channels of audio for left and right ears through

$$(y^l(n), y^r(n)) = f(x(n), p), \tag{1}$$

where $y^l$ and $y^r \in R^N$, and $f$ indicates the transformation function parameterized by the proposed framework. We denote the average of the binaural audio as $\bar{y} = \text{mean}(y^l, y^r)$.

Specifically, the relative position can be described with a tuple $p = (p_s, p_\alpha)$, where $p_s = (p_x, p_y, p_z)$ is the spatial position that indicated by the coordinates, and $p_\alpha = (qx, qy, qz, qw)$ is quaternions that indicates the head orientation from the listener to the source. The listener is stationary and set in the origin of the coordinate system, and therefore $(p_x, p_y, p_z)$ axes indicate the front, right and up directions respectively. Besides, we also denote the spatial position of the left and right ears of the listener as $p_{lstn}^l$ and $p_{lstn}^r$ respectively.

A simple way to align the temporal differences between the left and right ears is geometric warping, a non-parametric method conditioned on the distance between the source and the listener:

$$\rho(n) = n - C \cdot \|p_{src}(n) - p_{lstn}(n)\|, \tag{2}$$

where $n$ indicates the current time-stamp, $C$ is a constant that calculated by the ratio between the audio sampling rate and the speed of the sound. As the predicted warpfield $\rho(n)$ are usually floats, the warped signals can be computed with the linear interpolation:

$$x_{warp}(n) = (\lceil \rho(n) \rceil - \rho(n)) \cdot x_{\lfloor \rho(n) \rfloor} + (\rho(n) - \lfloor \rho(n) \rfloor) \cdot x_{\lceil \rho(n) \rceil}, \tag{3}$$

where $\lceil \cdot \rceil$ and $\lfloor \cdot \rfloor$ indicate the ceil and floor function respectively. The warping for the left and right ears can be obtained by changing the position $p_{lstn}(n)$, and the warped binaural audio is denoted as $(x_{warp}^l, x_{warp}^r)$, which are with low quality without considering the diffraction of audio.

### 3.2 Overview of the Framework

The proposed framework contains two stages, and we train two distinct diffusion models for each stage, parameterized by $\theta_c$ and $\theta_s$ respectively. In the first stage, the model $\theta_c$ aims to synthesize the common information of the binaural audio supervised by $\bar{y}$, i.e., we treat the mono average of two channels as the golden supervision of the common information. And in the second stage, conditioned

on the synthesized audio $y_c$ of the first stage, a two-channel diffusion model $\theta_s$ tries to synthesize the golden binaural audio $y = (y^l, y^r)$. We provide an illustration of the proposed framework in Figure 2. We will first introduce the backbone diffusion model and then the details of two stages as below.

### 3.2.1 Conditional Diffusion Models

Diffusion models are score-based generative models [12, 34, 8]. They are composed of a *forward process* and a *reverse process*. In the training stage, the forward process gradually destroys the data samples into Gaussian noise with a large number of time steps. At each time step $t$, the model learns a score function that denotes the gradient information. Then, in the reverse process, with learned score functions at predefined inference noise schedules, the model can generate clean data samples from Gaussian noise in an iterative denoising process.

The forward process converts the data samples $z_0$ into the isotropic Gaussian noise $\epsilon \sim \mathcal{N}(0, I)$ with a predefined variance schedule $0 < \beta_1 < \cdots < \beta_t < \cdots < \beta_T < 1$. The latent representations at time step $t$ can be directly calculated with:

$$q(z_t|z_0) = \mathcal{N}(z_t; \sqrt{\bar{\alpha}_t}z_0, (1 - \bar{\alpha}_t)\epsilon), \tag{4}$$

where the $\alpha_t := 1 - \beta_t$, and $\bar{\alpha}_t := \prod_{s=1}^t \alpha_s$ can denote a corresponding noise level at time step $t$.

In sampling, the reverse process starts from the isotropic Gaussian noise $p(z_T) \sim \mathcal{N}(0, I)$, and iteratively denoises the generated samples toward clean data $z_0$ with the conditioning information $c$ :

$$p_\theta(z_0, \cdots, z_{T-1}|z_T, c) = \prod_{t=1}^T p_\theta(z_{t-1}|z_t, c). \tag{5}$$

Providing strong conditioning information $c$ is usually helpful to reduce the number of inference steps and improve the generation quality. The Gaussian transition probability of each inference step is parameterized as:

$$p_\theta(z_{t-1}|z_t) = \mathcal{N}(z_{t-1}, \mu_\theta(z_t, t, c), \sigma_\theta^2 I), \tag{6}$$

where the variance $\sigma_\theta^2$ is usually predefined as $\frac{1-\bar{\alpha}_{t-1}}{1-\bar{\alpha}_t}\beta_t$ or $\beta_t$. Following previous works [12, 15], we use the former one in this work. The model is trained by maximizing the variation lower bound of the likelihood $p_\theta(z_0)$. When we parameterize the mean function as:

$$\mu_\theta(z_t, t, c) = 1/\sqrt{\alpha_t}(z_t - \beta_t/\sqrt{1 - \bar{\alpha}_t}\epsilon_\theta(z_t, t, c)), \tag{7}$$

a reweighted training objective is usually adopted in practice as [12, 4, 15]:

$$L_D(\theta) = \mathbb{E}_{z_0, \epsilon, t} \left\| \epsilon - \epsilon_\theta(\sqrt{\bar{\alpha}_t}z_0 + \sqrt{1 - \bar{\alpha}_t}\epsilon, t, c) \right\|_2^2. \tag{8}$$

For human speech signals, they usually have more energy in low-frequency region, while Gaussian noise has equal intensity at different frequencies. Hence, in the forward process, the high-frequency information is firstly destroyed, then the low-frequency information is also destroyed when the time step $t$ increases. Correspondingly, in the reverse process, the low-frequency information will be gradually generated at first, then details will be iteratively refined with following inference steps, which is suitable for accurate and high-fidelity speech waveform generation.

### 3.2.2 Details of Two Stages

With the understanding of the iterative refinement mechanism of diffusion models, we integrate them into the proposed two-stage framework, in which a single-channel diffusion model generates the common information of binaural audio in the first stage, while a two-channel diffusion model is then used for modelling the specific information of both left ear and right ear.

In the first stage, the average $\bar{y}$ of the golden binaural audio is considered as the clean data in the forward process. In this way, the diffusion model is encouraged to generate the common information of the two channels. The condition information is consistent for the forward and reverse processes. In concrete, the model takes the average of the warped binaural audio $\bar{x} = \text{mean}(x_{warp}^l, x_{warp}^r)$ as the condition information (Defined in Equation 3). In addition to the mono audio and position, the

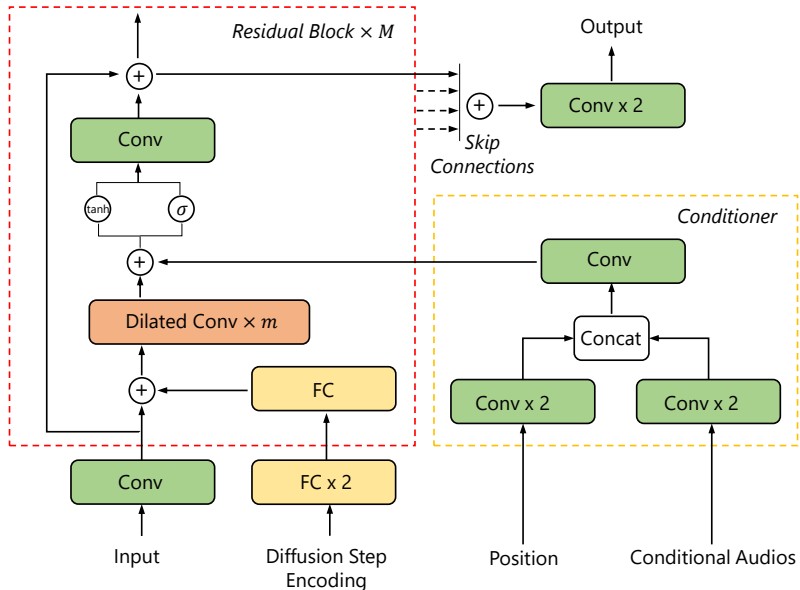

Figure 3: An illustration of the model architecture. "FC" indicates the fully connected layer, "Conv" indicates the $1 \times 1$ convolution layer, and "Dilated Conv" indicates the bidirectional dilated convolution layer. The input represents the corrupted data sample $z_t$ at the $t$-th diffusion step, while the position and the conditional audio have been defined in Section 3.1. We omit the activation functions for simplicity.

condition of the first stage can be denoted as $c_1 = (p_s, p_\alpha, \bar{x}, x)$. The position and audio information are separately processed by the model, which will be detailed introduced in Section 3.3.

The second stage aims at synthesizing the binaural audio through a two-channel diffusion model, therefore the golden audio $y = (y^l, y^r)$ is taken as the clean data for the two channels respectively. Different from that in the first stage, the condition information is distinct in the forward and reverse process as it includes the generation results of the first stage. The average $\bar{y}$ of the golden binaural audio is known in the forward process, therefore, along with the warped binaural audio, the condition information can be written as $c_2 = (p_s, p_\alpha, x^l_{warp}, x^r_{warp}, x, \bar{y})$. For the reverse process, the generation results $y_c$ of the first stage is utilized to replace the golden audio which is unknown in inference, i.e., $c_2 = (p_s, p_\alpha, x^l_{warp}, x^r_{warp}, x, y_c)$.

While training, the two stages are trained separately by minimizing the regression loss defined in Equation 8 utilizing different clean data and conditions. And in inference, firstly the common stage generates the mono audio $y_c$, conditioned on which the specific stage synthesizes the binaural audio, which are the final output of the framework. There exists an inconsistency between the training and inference of the second stage due to the different conditions, and fine-tune the second stage conditioned on the output of the first stage may alleviate this problem. We leave it for future work.

### 3.3 Model Architecture

The model architecture of our diffusion model is shown in Figure 3. The diffusion models utilized in the two stages have similar architectures, except the channels of the input, output and condition are one and two for the common and specific stages respectively. We introduce a Conditioner component to deal with the condition signals introduced in previous sections. Specifically, we utilize two separate convolution blocks to process the position as well as the conditional audio respectively, and the output features are concatenated and then processed by a final convolution layer. The conditional audio represents the concatenation of $[\bar{x}, x]$ and $[x^l_{warp}, x^r_{warp}, x, y_c]$ for the two stages. The output of the Conditioner is taken as the representation of condition signals.

The rest of our model generally follows the design in [15]. The main network is based on the bidirectional variant of the dilated convolution layer [24] given the non-autoregressive nature of the task. The architecture is constructed by $M$ residual blocks, with $m$ dilated convolution layers in

each block. The diffusion step $t$ is represented by positional encoding [40] and encoded by two fully connected layers before fed to the network, and then in each block, the diffusion step will be further transformed by another fully connected layer that is unique among blocks, where the output representation will be added to the input of the dilated convolution layer. The conditional representation modeled by the Conditioner will be added to the output of the dilated layer to bring in the conditional information. In the end, through skip connections, the outputs of residual blocks are gathered to produce the final output of the model.

## 4 Experiments

**Dataset**    We utilize the dataset released in [29][5] as it is the largest binaural dataset captured in the wild up to now. It contains 2 hours of mono-binaural parallel audio data at 48kHz from eight different objects, recorded in a regular room. The position and orientation between the objects and the listener are tracked at 120Hz and aligned with the audio. We follow the original split of training/valid/test sets to make our results comparable.

**Baselines**    The proposed framework is denoted as **BinauralGrad**, and we consider following baselines. **DSP** is a binaural rendering approach that simulates the spatial acoustic effects of sound sources with dynamic virtual positions. DSP utilizes RIR to model the room reverberance and HRTF to model the acoustical influence of the human head. We perform RIR simulation with an open-sourced tool[6] according to the room information provided in the original paper [29]. The HRTF data comes from [9], which was measured at a distance of 1.4 meters using the KEMAR, a manikin for acoustic testing. And we follow the procedure in [7] to simulate the binaural sound. As the exact HRTF and RIR are not provided in the dataset, therefore the DSP results are worse than that reported in the original paper [29]. **WaveNet** [24] where the positions of source and listener are used as condition signals and fed into a ConvNet to generate binaural audio from the mono one. **WarpNet** [29] which originally proposes the dataset, and stacking a neural time warping module with a temporal ConvNet to generate binaural audio. For all baselines, we adjust their hyper-parameters to make the number of parameters comparable with our model to ensure fair comparisons.

**Evaluation Metrics**    We use following metrics to evaluate the quality of synthesized binaural audio. **Wave L2**, which is the mean squared error between the synthesized binaural audio and the golden binaural recording. **Amplitude L2 and Phase L2**, which are the mean squared errors between the synthesized binaural speech and the binaural recording on the amplitude and phase respectively after performing Short-Time Fourier Transform (STFT) on wave. **PESQ**, which is the perceptual evaluation of speech quality[8] and widely used in speech enhancement. **MRSTFT**, which models the multi-resolution spectral loss[9] by taking the spectral convergence, log magnitude loss and linear magnitude loss into consideration. Except PESQ which is the higher the better, the rest of metrics are the lower the better. Besides object metrics, we also conduct subject evaluation (Mean Option Score, **MOS**) to verify the audio quality intuitively.

**Training Configurations**    As described in Section 3.3, BinauralGrad consists of $M = 3$ residual blocks, each of which has $m = 10$ bidirectional dilated convolution layers. The hidden size and the dimension of the diffusion step encoding are both set to $128$. We train BinauralGrad on 8 Nvidia V100 GPUs for 1M steps, and the diffusion steps are set to 200 and 6 during training and inference respectively, mainly following previous diffusion works [15].

We will introduce the experimental results in the rest of this section. We first compare the proposed BinauralGrad with baseline systems in terms of objective metrics, then verify the effectiveness of the proposed two-stage framework by comparing with single stage diffusion model. In addition, a subject metric MOS test is also utilized to show the intuitive performance of the model. Finally, to better understand the two-stage model, we also conduct a thorough analysis on the output of each stage.

---

[5]https://github.com/facebookresearch/BinauralSpeechSynthesis
[6]https://github.com/sunits/rir_simulator_python
[7]http://spatialaudio.net/ssr
[8]https://github.com/aliutkus/speechmetrics
[9]https://github.com/csteinmetz1/auraloss

Table 1: The comparison regarding binaural audio synthesis quality. For all baselines, we report the results based on our re-implementation.

| Model | Wave L2 ($\times 10^{-3}$) ↓ | Amplitude L2 ↓ | Phase L2 ↓ | PESQ ↑ | MRSTFT ↓ |
|---|---|---|---|---|---|
| DSP | 1.543 | 0.097 | 1.596 | 1.610 | 2.750 |
| WaveNet [24] | 0.179 | 0.037 | 0.968 | 2.305 | 1.915 |
| WarpNet [29] | 0.157 | 0.038 | 0.838 | 2.360 | 1.774 |
| BinauralGrad | **0.128** | **0.030** | **0.837** | **2.759** | **1.278** |

Table 2: The comparison with the single stage diffusion model.

| Model | Wave L2 ($\times 10^{-3}$) ↓ | Amplitude L2 ↓ | Phase L2 ↓ | PESQ ↑ | MRSTFT ↓ |
|---|---|---|---|---|---|
| Single Stage | 0.144 | 0.031 | 0.877 | 2.544 | **1.269** |
| BinauralGrad | **0.128** | **0.030** | **0.837** | **2.759** | 1.278 |

## 4.1 Main Results

We report the object metrics of BinauralGrad and other binaural speech synthesis baselines in Table 1. The proposed framework performs consistently better than compared baselines over different metrics. Specifically, we have several observations:

- Our method outperforms the main baseline WarpNet on Wave L2 by a large margin, showing that BinauralGrad can synthesize better binaural waveforms that are closer to the golden recordings.

- Since there are some randomness in the waveform, the metrics on the STFT results of waveform is important to evaluate the overall quality of binaural speech synthesis. Our method surpasses baseline methods by a large margin in terms of Amplitude L2 and MRSTFT, and being slight better than WarpNet on Phase L2.

- PESQ is a widely used metric on the perceptual evaluation of speech quality. The proposed BinauralGrad achieves 2.759 PESQ score, which is significantly better than other baseline systems. The naturalness of our method is further verified by the MOS results listed in Table 3.

In conclusion, the object metrics show that the proposed two-stage framework is able to generate more accurate binaural audio than existing baselines.

## 4.2 Comparison with Single Stage Diffusion Model

We compare BinauralGrad with a single stage diffusion model to verify the advantage of the two-stage framework. For the single stage diffusion model, its model architecture is similar to the two-channel diffusion model in the second stage of BinauralGrad, except for two differences: 1) the single stage model does not use any information from the common information, i.e., with condition $c = (p_s, p_\alpha, x^l_{warp}, x^r_{warp}, x)$; 2) the single stage model is two times deeper so as to make the number of parameters comparable.

The comparison of BinauralGrad with the single stage diffusion model are shown in Table 2. The results show that the two-stage model performs better than the single stage one on most metrics including PESQ and L2 losses of waveform, amplitude and phase, while achieving comparable MRSTFT scores, demonstrating the effectiveness of two stage framework. It is worth noting that the single stage model already outperforms the major baseline WarpNet over most metrics except Phase L2 as shown in Table 1, illustrating the stronger ability of diffusion models in synthesizing raw audio waveform than purely convolution based models.

## 4.3 MOS Test

We conduct three types of MOS to verify the quality of synthesized audio with human evaluation, including: 1) MOS, where judges are asked to rate for the overall naturalness and fluency of synthesized audio; 2) Similarity MOS, where judges are asked to rate for the similarity between the synthesized results and the golden binaural recordings; 3) Spatial MOS, where the judges are asked

Table 3: The MOS test results on the synthesized binaural audio. Best scores over baselines are marked bold.

| Model | MOS | Similarity MOS | Spatial MOS |
|---|---|---|---|
| Recording | $3.69 \pm 0.24$ | $4.52 \pm 0.23$ | $4.01 \pm 0.13$ |
| DSP | $3.41 \pm 0.23$ | $3.97 \pm 0.37$ | $3.75 \pm 0.15$ |
| WaveNet [24] | $3.37 \pm 0.20$ | $4.14 \pm 0.29$ | $3.41 \pm 0.15$ |
| WarpNet [29] | $3.61 \pm 0.24$ | $4.25 \pm 0.25$ | $3.77 \pm 0.13$ |
| BinauralGrad | $\mathbf{3.80} \pm 0.23$ | $\mathbf{4.43} \pm 0.20$ | $\mathbf{3.79} \pm 0.16$ |

Table 4: Analysis of BinauralGrad. Stage 1 stands for the first/common stage and stage 2 stands for the second/specific stage in BinauralGrad.

| ID | Setting | Wave L2 ($\times 10^{-3}$) $\downarrow$ | Amplitude L2 $\downarrow$ | Phase L2 $\downarrow$ | PESQ $\uparrow$ | MRSTFT $\downarrow$ |
|---|---|---|---|---|---|---|
| 1 | Mono | 1.340 | 0.063 | 1.564 | 1.410 | 2.159 |
| 2 | Physical Warp | 0.725 | 0.060 | 1.584 | 1.411 | 2.140 |
| 3 | Stage 1 | 0.300 | 0.043 | 1.070 | 2.000 | 1.682 |
| 4 | $(\bar{y}, \bar{y})$ | 0.200 | 0.031 | 0.598 | 3.031 | 1.020 |
| 5 | Stage 2 | 0.128 | 0.030 | 0.837 | 2.759 | 1.278 |
| 6 | Stage 2 on ID 4 | 0.013 | 0.013 | 0.372 | 4.212 | 0.697 |

to rate for the sense of direction contained in the synthesized audio. For all MOS tests, the judges rate discrete score from 1 to 5, and the higher score the better quality. The MOS results together with the 95% confidence interval are shown in Table 3.

Firstly, the proposed BinauralGrad achieves a quite high MOS score $3.80$ which not only surpasses all baselines, but also slightly outperforms the recording, illustrating the strong ability of the proposed framework in synthesizing natural audio waveforms. Secondly, the similarity MOS shows that BinauralGrad can synthesize closest binaural audio to the golden binaural recording among all compared models, which is consistent with the conclusions from objective metrics. Finally, the results on the spatial MOS show that DSP, WarpNet and BinauralGrad achieve similar performance on describing the sense of spatial, which might be attributed to the fact that the above three methods all utilize the physical warping process that brings in the interaural time difference (ITD) and results in the sense of direction [43].

## 4.4 Analysis

In this section, we conduct a thorough analysis to verify the effectiveness of each stage in our framework, and try to find out the potential bottleneck that brings sparks for future work.

In each stage, the diffusion model synthesize audio conditioned on different information. Therefore, to verify the effectiveness of the model in each stage, we evaluate the performance of the condition and synthesized audio separately to illustrate the improvements brought by the model. Specifically, in addition to the synthesized audio from the two stages (ID 3 and 5), we also calculate the performance of the physical warping results (i.e., $(x^l_{warp}, x^r_{warp})$, ID 2) and the average $\bar{y}$ of the golden binaural audio. Note that as the output of the first stage and $\bar{y}$ are both mono audio, we therefore calculate the scores by duplicating them to binaural such as $(\bar{y}, \bar{y})$ for ID 4 and $(y_c, y_c)$ for ID 3.

The results are listed in Table 4. Firstly, we can find that the performance of the physical warping (ID 2) is worse as it only brings gains on Wave L2 and remain consistent on other metrics compared to the mono audio (ID 1). Secondly, the improvements from ID 2 to 3 and 3 to 5 are brought by the first and second stage of the proposed model respectively, verifying the effectiveness and necessities of both stages.

In addition, we try to explore the boundary of the framework by directly feeding the perfect condition to the second stage while inference, i.e., instead of conditioning on $y_c$ which is synthesized by the first stage, we utilize the golden average $\bar{y}$ as the condition which is unavailable in inference to test the best performance of the model. As a result, we can find that the second stage achieves an extremely good result and brings a large improvement over all metrics compared with the given condition,

showing that the model in the second stage has been well trained. On the contrary, there still exists a large margin between the synthesized result from the first stage and the label (ID 3 vs. ID 4). In conclusion, the results in Table 4 show that the first stage is the bottleneck of the framework, which points out potential future work directions such as the end-to-end optimization of the two stages.

## 5  Conclusion

In this paper, we propose BinauralGrad, a two-stage framework for binaural audio synthesis conditioned on mono audio. Specifically, we formulate the synthesis process from a novel perspective and divide the binaural audio into a common part that is shared by two channels as well as distinct parts. Accordingly, a single-channel diffusion model is utilized to generate the common information in the first stage, conditioned on which a two-channel diffusion model synthesizes the binaural audio. The proposed framework is able to synthesize accurate and high-fidelity audio samples. On a benchmark dataset, BinauralGrad achieves state-of-the-art results both in object and subject evaluation metrics. In the future, we plan to improve the training of the two stages in an end-to-end way for better performance or speed up the inference of BinauralGrad, which is important for the online deployment of our method. The negative impact of our method might be the abuse, e.g., generating fake binaural audio to mislead the user in some VR-based human-computer interactive games, resulting in physical injury.

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
