# OpenReview forum: "BinauralGrad: A Two-Stage Conditional Diffusion Probabilistic Model for Binaural Audio Synthesis"
_NeurIPS.cc/2022/Conference — NeurIPS 2022 Accept_

### Official Review · Reviewer_6RDx · 2022-07-09

**Rating:** 7
**Confidence:** 5
**Soundness:** 3 good
**Presentation:** 4 excellent
**Contribution:** 3 good

**Summary:**

This is a paper that applies the diffusion model to a binaural audio generation task. The authors use a two-stage method to accomplish their task and use a modified wavenet-like structure as their model architecture. After thorough and detailed experiments, the authors achieve the best results among the listed models, which nicely support their conclusions. The contributions of the paper include:
- A new two-stage framework is used to generate high-quality binaural audio, with the assistance of a diffusion model.
- The experiment results show that the proposed method achieves excellent performance on this task.




**Questions:**

- The authors mention that they use warped binaural audio as conditional input, but never explain how this audio is warped. We would like to know if these audios are warped using traditional DSP methods, or other additional methods? Does the use of such conditional inputs lead to some kind of information leakage, and have the authors performed ablation experiments on this?
- The authors use the pattern of frequency destroying they found in the forward process of the diffusion model to explain why they use a two-stage generation method. However, in Table 2, the frequency-related evaluation metric MRSTFT of the proposed method is not as good as that of the single-stage model. Can the authors explain this phenomenon?
- Have the authors compared the generation speed of single-stage and two-stage models and is there a significant difference between the two?




**Limitations:**

The authors make a good statement about the limitations of their work.


**Strengths And Weaknesses:**

Strengths:
- This paper is easy to follow. The authors give a detailed background of the task, a clear definition of the problem, and a comprehensive explanation of the proposed method.
- The experiments are adequate. The authors have verified the feasibility of their method through comprehensive experiments and subjective and objective evaluations, which support their conclusions very well.


Weaknesses:
- The values of the standard deviations for all results in Table 3 seem to be a bit too large, which indicates that there may be a large disagreement among raters about the experimental results. Such results may be unconvincing.

Besides that, there are no significant weaknesses in this paper, but there are some issues that are difficult to explain and could be potential weaknesses, which will be mentioned in the next section.

---

> ### Author Response · Authors · 2022-08-02
> **Response to Reviewer 6RDx**
>
> We sincerely thank for your time and effort on reviewing our paper. We are grateful for your positive feedback. Our paper benefits a lot from your valuable and constructive comments. We response to your questions as follow.
>
> **Q1: About the standard deviations of MOS test.**
>
> Since binaural audio provides strong spatial cues, it is more complex for subjective evaluation of quality than the mono audio in text to speech (TTS). Hence, the MOS test on binaural audio might have a larger variance. From our literature review, the variance of MOS in binaural audio synthesis [1] is typically higher than that of TTS [2].
>
> **Q2: About the warped binaural audio.**
>
> We use the geometric warping mentioned in preliminary in Section 3.1 as warped binaural audio. We make this more clear in the revised version of our paper.
>
> The geometric warping is a non-parametric method, the physics law behind the geometric warping is that it will takes a period of time (distance divided by the speed of sound) for the sound to emit from source to listener. So geometric warping is to delay the mono audio by that period of time, leveraging only the source mono audio and the position of source and listener, which means there is not any information leakage.
>
> Moreover, since geometric warping reflects the physics process of sound emission, using geometric warping will help the binaural synthesizer to better align the temporal differences of input and the target, which is a common practice in binaural audio synthesis and also used in our baseline systems [1].
>
> **Q3: About the pattern of frequency destroying.**
>
> We are sorry for the ambiguous description. The analysis for the pattern of frequency destroying is our observation of diffusion models for speech signals (please refer to Q2 of [Response to Reviewer BZFH](https://openreview.net/forum?id=_FMJmDEPLzs&noteId=CX-Ltx7XdaZ) for experimental details), which is the motivation of using diffusion models instead of the motivation of proposing the two-stage framework. To clearly describe our ideas, we move the description for the pattern of frequency destroying from the beginning of section 3.2.2 to the end of section 3.2.1 in the revised version of our paper.
>
> For binaural audios, as human leverages the interaural time and level differences (ITD and ILD) [3] for a sense of direction, the waveform metrics are important. Hence, we directly conduct our experiments in time domain instead of frequency domain. Apart from Wave L2, PESQ is commonly used for evaluating speech quality in time domain. As our proposed two-stage framework outperforms the single-stage model and the baseline models in these metrics, we think that the effectiveness of the framework for binaural audio generation can be verified.
>
> **Q4: About the difference of single-stage with two-stage model on latency.**
>
> In our test, there is no significant difference between the generation speed of single-stage model and two-stage model. As we mention in section 4.2, we use the single-stage model that two times deeper than our two-channel diffusion model in the second stage of BinauralGrad, in order to make the model size comparable. During the inference process, as the two-stage framework spends more time on loading models, the latency of two-stage model will be slightly higher than the single-stage model.
>
>
>
> **Reference**
>
> [1] Alexander Richard, Dejan Markovic, Israel D Gebru, Steven Krenn, Gladstone Alexander Butler, Fernando Torre, and Yaser Sheikh. Neural synthesis of binaural speech from mono audio. In ICLR, 2021.
>
> [2] Yi Ren, Yangjun Ruan, Xu Tan, Tao Qin, Sheng Zhao, Zhou Zhao, and Tie-Yan Liu. Fastspeech: Fast, robust and controllable text to speech. Advances in Neural Information Processing Systems, 32, 2019.
>
> [3] Martin Raspaud, Harald Viste, and Gianpaolo Evangelista. Binaural source localization by joint estimation of ILD and ITD. IEEE Transactions on Audio, Speech, and Language Processing, 2010.

---

### Official Review · Reviewer_YLbm · 2022-07-11

**Rating:** 5
**Confidence:** 4
**Soundness:** 3 good
**Presentation:** 2 fair
**Contribution:** 2 fair

**Summary:**

This paper studies the problem of binaural audio synthesis, and proposes to decompose the binaural audio into a common part that shared by the left and right channels as well as a specific part that differs in each channel. A two-stage framework based on diffusion models is proposed to synthesize the two channels, respectively. Experiments on a benchmark dataset demonstrate the effectiveness of the proposed framework.

**Questions:**

Diffusion models are powerful and lead to good gains in terms of binaural sound synthesis results, it would be important to discuss why it is helpful and suitable for this specific task, compared to prior methods as well as its limitations.

**Limitations:**

No limitations are discussed in the paper. There is only one sentence that discusses the societal impact: The negative impact of our method might be the abuse of binaural speech synthesis, which is very vague and unclear.

**Strengths And Weaknesses:**

Strengths: The idea to use diffusion models for this task is interesting, and the proposed two-stage framework is well-motivated. Generally, the paper is clearly written and easy to understand. Table 1 clearly shows that the proposed method based on diffusion models compares favorably to prior methods including the latest neural method WarpNet [29]. The main contribution/strength of this paper is introducing a technique---diffusion model, which is demonstrated to be powerful in many other areas, to this existing task of binaural sound synthesis, and achieves some gains.

Weakness: The motivation for dividing the binaural synthesis problem into two stages can be better discussed. For example, in Figure 1, it would be useful to explicitly illustrate what information is used in the first stage and what information is used in the second stage, instead of just showing the waveforms. Moreover, diffusion models are powerful and lead to good gains in terms of binaural sound synthesis results, it would be important to discuss why it is helpful and suitable for this specific task, compared to prior methods as well as its limitations. No limitations or failure cases are currently discussed across the paper and only one sentence discusses potential societal impact.

---

> ### Author Response · Authors · 2022-08-02
> **Response to Reviewer YLbm**
>
> We sincerely thank for your time and effort on reviewing our paper. We are grateful for your positive feedback. Our paper benefits a lot from your valuable and constructive comments. We response to your questions as follow.
>
> **Q1:  About the motivation for two-stage model.**
>
> Considering that the generation of binaural audio consists of two steps: 1) the original mono audio is emitted by the source and then diffuses to the listener, 2) the binaural audio is then encoded by human head depending on the marginal difference of audio received at two ears. We propose to leverage two-stage model where the common stage models the fundamental information of the source mono audio and the specific stage focuses on modeling the difference of left ear and right ear.
>
> From the analysis in section 4.4 about Table 4, the improvement from ID 2 to 3 is the advantages of first (common) stage and the improvement from 3 to 5 is the advantages of second (specific) stage, which shows that both stages is necessary and non-trivial to optimize, verifying the effectiveness of two-stage design.
>
> **Q2:  About the improvement on Figure 1.**
>
> Since Figure 1 is used for a high-level illustration, we hope we can keep the simpleness of this figure and the detailed information of each stage is shown in Figure 2. We add some explanations to the caption of Figure 1 in the revised paper to give readers a better understanding of the two stages: 1) The common stage models the factor affecting both the left and right channel such as the distance of source and listener and the interaction of audio with room and head. 2) And the specific stage models the difference of left ear and right ear including the marginal distance difference between two ears and the difference of the impulse response of two ears.
>
> **Q3: About the helpfulness of diffusion models.**
>
> Since diffusion models have shown strong ability in modelling  high-dimension distribution of raw waveforms and can generate high-fidelity audio on the task of speech synthesis with their iterative refinement mechanism, we utilize diffusion model for binaural audio synthesis. Moreover, we expect to highlight that our contribution is two-fold. The first one is proposing a two-stage framework based on the physical characteristics to improve the quality of synthesized binaural audio, while the second one is employing diffusion model in this framework to achieve the strong results.
>
> **Q4: About the limitation and negative impact.**
>
> For limitation, we add the discussion on the latency of our method in the conclusion of revised paper, which is also a common shortcoming of diffusion models and can be an future research direction (as mentioned in Q1 of [Response to Reviewer BZFH](https://openreview.net/forum?id=_FMJmDEPLzs&noteId=CX-Ltx7XdaZ)). With more application of binaural audio on VR or Metaverse in the future, the negative impact of our method might be the abuse, e.g., generating fake binaural audio to mislead the user in some VR-based human-computer interactive games, resulting in physical injury.

---

### Official Review · Reviewer_BZFH · 2022-07-11

**Rating:** 8
**Confidence:** 4
**Soundness:** 4 excellent
**Presentation:** 3 good
**Contribution:** 3 good

**Summary:**

In this paper, the authors propose a diffusion-based system that can directly generate binaural signals. Based on analysis of the way sound propagates and humans perceive spatial impressions of sound, the authors designed a two-step system. The proposed system is evaluated by both objective measures and subjective evaluations, both of which shows that the proposed system is trained successfully.

**Questions:**

L162-163: Just curious, Why? White noise has equal energies all over the frequency.
Is it bc the test signals have, due to our auditory perception, more energy on
the low-frequency region? It could be useful to discuss this

In general: There are many symbols (x, y, with bars and hats etc). I found it quite confusing at the beginning and once misunderstood something was wrong.


**Limitations:**

(Mentioned in the Weakness)

**Strengths And Weaknesses:**

Strength: The proposed system is designed based on a nice intuition. It’s novel, and effective. The design and result of the experiment are solid. The writing is good, too.

Weakness: Directly synthesizing the binaural signals is not practical as of now given its high computation complexity. (But I do not think this is not a critical weakness)

---

> ### Author Response · Authors · 2022-08-02
> **Response to Reviewer BZFH**
>
> We sincerely thank for your time and effort on reviewing our paper. We are grateful for your positive feedback. Our paper benefits a lot from your valuable and constructive comments. We response to your questions as follow.
>
> **Q1: Directly synthesizing the binaural signals is not practical as of now given its high computation complexity.**
>
> The latency is a common issue of diffusion-based models as they leverage multiple denoising steps to achieve high generation quality, while the proposed BinauralGrad can synthesize binaural audio (48kHz sampling rate) in only 6 steps for each stage with the mono audio serving as a strong condition signal. For diffusion models sampling in high dimensional space, this has been a small number of inference steps. BinauralGrad is suitable for offline scenario such as the synthesis of binaural recording in movies. And it will be an interesting future work for BinauralGrad to further reduce inference step based on the task characteristic. For instance, we may extract a more informative prior for the noise distribution used in diffusion models from the mono audio to speed up the inference speed.
>
> **Q2: About the frequency destruction.**
>
> Yes, it is because human speech signals usually have more energy in the low-frequency region.
> Energy can be shown by signal amplitude in time domain. In our early-stage experiments, we decompose human speech recordings $x_{0}$ into $N$ parts in time domain with a sampling rate of $24$kHz according to the frequency subband, by utilizing the PQMF algorithm which is widely used in literature[1][2], and different PQMF filters have been tested. Firstly, we observe that the signals in the low-frequency band, e.g., from $0$kHz to $6$kHz, distinctively have larger amplitude than that in the high-frequency band, e.g., from $18$kHz to $24$kHz. Secondly, we compute the destroyed signals $x_{t}$ according to the forward process of DDPM with $T=1000$. When $t$ is small, e.g., $t=200$, we observe that some high-frequency parts have been destroyed while the low-frequency part still has semantic information. When $t$ becomes large, e.g., $t=600$, the low-frequency part is also destroyed. When $t=1000$, the signals in each frequency band are transformed into Gaussian noise.
>
> By the way, considering the comments from Reviewer 6RDx, we move this description from the beginning of section 3.2.2 to the end of section 3.2.1 in the revised version of our paper.
>
> **Q3: About the confusing symbols.**
>
> We are sorry for the confusing symbols. $\hat{x}$ stands for the physical warp of mono audio defined in Equation (3), and we change it to $x_{warp}$ in the revised paper to make it more distinguishable.
>
>
> **References**
>
> [1] Geng Yang, Shan Yang, Kai Liu, Peng Fang, Wei Chen, and Lei Xie. Multi-band melgan: Faster waveform generation for high-quality text-to-speech. In SLT, 2021.
>
> [2] Chengzhu Yu, Heng Lu, Na Hu, Meng Yu, Chao Weng, Kun Xu, Peng Liu, Deyi Tuo, Shiyin Kang, Guangzhi Lei, Dan Su, and Dong Yu. Durian: Duration informed attention network for speech synthesis. In INTERSPEECH, 2020.

---

### Meta-Review · Area_Chair_5fm9 · 2022-08-26

**Recommendation:** Accept
**Confidence:** Certain

**Metareview:**

The use of diffusion models for binaural audio synthesis is interesting and the two-stage design is novel and well motivated. The authors also addressed the reviewers concerns.

**Award:**

No

---

### Decision · Program_Chairs · 2022-09-14

Accept